

# Investigating compound flooding in an estuary using hydrodynamic modelling: A case study from the Shoalhaven River, Australia.

Kristian Kumbier[1,2], Rafael C. Carvalho[2], Athanasios T. Vafeidis[1], Colin D. Woodroffe[2]

[1]Department of Geography, University of Kiel, Kiel, 24113, Germany

5  [2]School of Earth and Environmental Sciences, University of Wollongong, Wollongong, 2500, Australia

*Correspondence to*: Kristian Kumbier (kkumbier@uow.edu.au)

**Abstract.** Previous modelling studies have considered storm-tide and riverine flooding independently, even though joint-probability analysis highlighted significant dependence between extreme rainfall and extreme storm surges in estuarine environments. This study investigates compound flooding by quantifying horizontal and vertical differences in coastal flood risk estimates resulting from a separation of storm-tide and riverine flooding processes. We used an open source version of the Delft3D model to simulate flood extent and inundation depth due to a storm event that occurred in June 2016 in the Shoalhaven Estuary, southeast Australia. Time series of observed water levels and discharge measurements are used to force model boundaries, whereas observational data such as satellite imagery, aerial photographs, tidal gauges and water level logger measurements are used to validate modelling results. The comparison of simulation results including and excluding riverine discharge demonstrated large differences in modelled flood extents and inundation depths. A flood risk assessment accounting only for storm-tide flooding would have underestimated the flood extent of the June 2016 storm event by of 30 % (20.5 km²). Furthermore, inundation depths would have been underestimated on average by 0.34 m and by up to 1.5 m locally. We recommend to consider storm-tide and riverine flooding processes jointly in estuaries with large catchment areas, which are known to have a quick response time to extreme rainfall. In addition, comparison of different entrance conditions indicated that permanently opening the intermittent entrance, in order to reduce exposure to riverine flooding, would increase tidal range and exposure to both storm-tide flooding and wave action.

## 1 Introduction

Storm surges are the main driver of coastal flooding leading to loss of human life, destruction of homes and civil infrastructure, and disruption of trade, fisheries and industry (Resio and Westerink, 2008). An increase in sea-level is expected to exacerbate storm surge related risks to coastal communities, because the frequency and extent of coastal flooding is likely to increase (IPCC, 2014; Vitousek et al., 2017). The impacts of a storm surge may further intensify when it coincides with high spring tide (Pugh, 2004) and/or riverine flooding (Zheng et al., 2013). Extreme water levels resulting from a combination of storm-tide flooding and riverine flooding are also known as coincident or compound flood events (IPCC, 2014; Leonard et al., 2014). For some time the two flooding drivers involved were treated independently in coastal



flood risk assessments (Torres et al. 2015), even though joint-probability analysis highlighted significant dependence between extreme rainfall and extreme storm surges (Svensson and Jones, 2004, 2006; Zheng et al., 2013, Zheng et al., 2014). Therefore studies such as those of Svensson and Jones (2004, 2006) have suggested to take both processes into account for flood risk estimations in coastal areas.

Estuarine and deltaic environments are particularly at risk from compound flooding due to their exposure to storm-tides and riverine discharges (Olbert et al., 2017).   Studies on compound flooding have been assessed on local scales using hydrodynamic models and joint-probability statistics (Lian et al., 2013; Chen et al. 2014; Olbert et al., 2017); and on regional to national scales using joint-probability statistics (Zheng et al., 2013; Wahl et al., 2015). Olbert et al. (2017) have investigated the interaction of storm-tide and riverine flooding drivers in the Lee estuary using a coupled ocean-

hydrodynamic model. Their detailed analysis for Cork city revealed a primarily fluvial driven flooding regime that is enhanced by storm-tide water levels. Chen et al. (2014) explored compound flooding in the Tsengwen River using a three-dimensional hydrodynamic model. Their modelling approach included separate analysis of tide and riverine flooding drivers that were ultimately combined to assess differences in flood extent and depth. Lian et al. (2013) used a combination of joint-probability statistics and hydrodynamic modelling techniques to investigate the flood severity resulting from compound

flooding for the city of Fuzhou. Statistical analysis on compound flooding in the United States showed on one hand an increasing risk of compound flooding for major US cities, and on the other hand, the need for more research on local scale to quantify the actual impacts associated with compound events (Wahl et al., 2015). Svensson and Jones (2004) examined the dependence of the two involved processes in estuaries around Great Britain. They identified estuaries with steep catchments as prone to combined storm-tide and riverine flooding, because of the quick catchment respond to abundant rainfall.

The southeast coast of Australia features more than 120 estuaries (Roy et al. 2001), with some of them being characterized by steep catchments. Thereby, the close proximity of the Great Escarpment to the Pacific Ocean may promote compound flooding in those estuaries that are known to have quick response times to extreme rainfall (Nanson and Hean 1985). Zheng et al. (2013) observed statistically significant dependence between extreme rainfall and storm surge residuals along the east coast of Australia and advised to consider both processes jointly to correctly quantify flood risk. Uncertainties in flood risk

estimations can result from horizontal (flood extent) and vertical dimensions (inundation depth).

A key component of any flood risk assessment is the preparation of flood maps, which aim to identify coastal areas threatened by flooding. This is usually done through static or dynamic modelling approaches. The most simple approach is the so-called static or bathtub modelling approach. The static model is based on the assumption that areas lower than a certain extreme water level are inundated if there is hydrological connectivity (Poulter and Halpin, 2008; Van de Sande et

al., 2012). Resulting flood maps are known to generally overestimate the flood extent due to the omission of important factors influencing floodwater flow such as bottom friction, the conservation of mass and flood duration (Bates et al., 2005; Gallien et al., 2011; Breilh et al., 2013; Ramirez et al., 2016; Seenath et al., 2016; Vousdouskas et al., 2016). Further limitations to modelling compound flooding using the static approach result from the restriction of input arguments. The approach allows only for a specific extreme water level as an input and not spatially varying water levels from different



flooding drivers such as those resulting from an incoming riverine flood wave and a storm-tide flood wave. The dynamic modelling approach utilizes a hydrodynamic model to simulate the flow of floodwater resulting from various sources such as storm-tides and riverine discharges. These models have been applied successfully in coastal flood risk assessments at different scales and with varying degrees of model complexity (Bates et al., 2005; Breilh et al., 2013; Gallien et al., 2011;

Skinner et al., 2015; Seenath et al., 2016; Ramirez et al. 2016, Vousdouskas et al., 2016). A comprehensive overview of different flood inundation modelling methods as well as recent developments can be found in Teng et al. (2017). Considering the potential flooding drivers that hydrodynamic models can account for, they are the more appropriate tool to assess the extent and depth of flooding resulting from compound flood events.

In this study we investigate a compound flood event in a southeast Australian estuary using an open source version of the

hydrodynamic model Delft3D. In June 2016, a storm surge coincided with extreme riverine discharge in the Shoalhaven Estuary. Time series of observed water levels and discharge measurements are used to force model boundaries, whereas observational data such as satellite imagery, aerial photography, tidal gauges and water level logger measurements are used to validate modelling results. By modelling the involved flooding drivers separately and jointly we quantify horizontal and vertical differences in flood risk estimation. Assessing these differences in flood extent and inundation depth reveals

potential uncertainties resulting from a separation of storm-tide and riverine processes in coastal flood risk assessments. Obtaining detailed insights into local scale compound flooding is of great relevance for future flood risk management of estuaries. This is particularly important for certain Australian estuaries, which have been shown to be subject to compound flood events (Zheng et al., 2013). Furthermore, we address the site specific influence of changing entrance conditions on modelled water levels and the extent of flooding. The Shoalhaven Estuary is characterized by two entrances: a permanently

open and an intermittent entrance. These are considered in the modelling through different open boundary setups.

The objectives of this study are:

1. To understand the interaction of storm-tide and fluvial flooding mechanisms by modelling a compound flooding event in
the estuary
2. To quantify horizontal and vertical modelling differences resulting from a separation of the involved flooding drivers
3. To quantify how changing entrance conditions affect modelled water levels and flood extent

## 2 Study Area

### 2.1 Geomorphological and hydrodynamic setting

The Shoalhaven River is located on the southeast coast of New South Wales (NSW), Australia (Fig. 1). The coastline is controlled mainly by waves; tides are semi-diurnal with a significant diurnal inequality. The lower Shoalhaven River is referred to as the Shoalhaven Estuary, which is classified as a mature barrier estuary with a catchment size of approximately



7150 km² (Roy et al., 2001). The discharge of the Shoalhaven River is regulated by Tallowa dam, which is located approximately 68 km upstream from the coast. The largest settlement within the floodplain is Nowra, which is located 18 km from the coast. The tidal range in Crookhaven Heads at the mouth of the estuary is about 1.5 m during spring tides. It decreases by 0.25 m further upstream towards Nowra, before the tidal range even slightly amplifies for several kilometres

5    (MHL, 2012). The annual mean rainfall of the Shoalhaven River catchment is approximately 900 mm per year (Carvalho and Woodroffe, 2015). In June 2016 a storm event caused extensive inundation of the floodplains surrounding the lower Shoalhaven River.

**Figure 1: Map showing the lower Shoalhaven River and tidal gauges (red dots). LiDAR derived topographic data of the floodplain**
10    **is presented in m AHD.**



The waterway of the Shoalhaven Estuary is quite unusual with a permanent opening at Crookhaven Heads and an intermittent entrance at Shoalhaven Heads (Fig. 1). This environmental setting of two entrances of different nature results from the construction of Berrys Canal by landowner Alexander Berry in 1822. Originally the estuary had its opening to the Pacific Ocean at Shoalhaven Heads, but with the construction of Berrys Canal the discharge has been redirected towards

Crookhaven Heads, which is more protected from wave action and permanently open. In consequence, Shoalhaven Heads turned into an intermittent opening, which only breaches during large storm events (Umitsu et al., 2001). Broughton Creek is the largest tributary in the northern part of the floodplain, whereas the southern part is drained by the much smaller Crookhaven River. The floodplain of the estuary is characterized by low-lying alluvial plains, which developed through estuarine infilling during the last 6000 years (Woodroffe et al., 2000). The majority of the floodplain is elevated between 1

and 2 m above Australian Height Datum (AHD), with some areas being even below zero (Fig. 1). The vertical datum AHD approximates mean sea-level.

## 2.2 The June 2016 storm event

The June 2016 storm event was due to an East Coast Low (ECL), which formed northeast of Queensland and tracked south along the eastern coastline of Australia. ECLs are low pressure cyclones, which form in certain synoptic situations initially

as a trough and move parallel to the coast of Queensland and northern NSW (Shand et al., 2011).

During the night of 5 to 6 June 2016 mean sea-level pressure dropped to a minimum of 991.5 mbar at nearby Port Kembla station (approximately 45 km north of the Shoalhaven Estuary), while maximum wind gusts of 27.7 m s$^{-1}$ from an easterly direction were measured (BOM, 2016). The storm generated a positive surge of 0.85 m at the estuary's entrance in Crookhaven Heads (Fig. 2). The non-tidal residual (NTR) estimation of 0.85 m during the storm event was calculated by

comparing the predicted astronomical tide to the observed water level and is explained in more detail in Sect. 3.1.





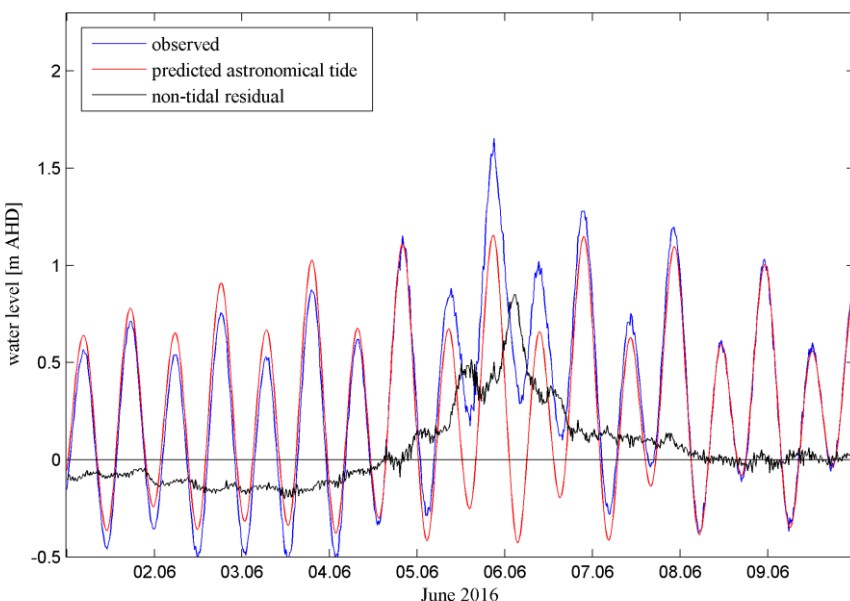

**Figure 2: Observed water level at Crookhaven Heads gauge (blue), predicted astronomical tide (red) and non-tidal residual (black) in m AHD, during the June 2016 storm event.**

The pattern of highest NTR during low water suggests non-linear tide-surge interactions, which are known to occur in

shallow waters due to changes in the phase speed of the tidal and surge wave (Flather, 2001; Horsburgh and Wilson, 2007). Therefore, it is more appropriate to express the surge height by the metric of skew surge, which is giving the absolute difference within a tidal cycle between the maximum observed water level and the predicted tidal high water, irrespective of time of occurrence (Williams et al., 2016). The skew surge of the June 2016 storm-tide was about 0.5 m at the entrance of the Shoalhaven Estuary. This measure is consistent with estimates of NTR for the June 2016 storm event reported by

Burston et al. (2016). Maximum wave heights of more than 9 m with a wave period of 15 s from an easterly direction were measured at Port Kembla buoy (BOM, 2016). The weekly cumulative rainfall measured for Wollongong (45 km north of the Shoalhaven River) was approximately 289 mm (Burston et al., 2016).

## 3. Data and methods

### 3.1 Model input data

The topographic dataset of the study area represents bare earth and originates from Light Detection and Ranging (LiDAR) measurements collected at national scale from 236 individual LiDAR surveys between 2001 and 2015. It can be downloaded from the server of Geoscience Australia (http://www.ga.gov.au/elvis/). The data has a spatial resolution of 5 m, a vertical accuracy of at least 0.3 m AHD (95 % confidence) and a horizontal accuracy of at least 0.8 m (95 % confidence). Bathymetric data of the Shoalhaven Estuary originate from 103.265 point measurements vertically referenced to AHD taken





during hydrographic surveys between September 2005 and November 2006 by the NSW Office of Environment and Heritage (OEH). The data is accessible from the OEH homepage (http://www.environment.nsw.gov.au/estuaries/stats/ShoalhavenRiver.htm). The bathymetric point measurements were interpolated to a raster surface of 5 m spatial resolution using an ordinary Kriging method with a spherical semivariogram

model. The accuracy of this interpolation was assessed following the method presented in Chaplot et al. (2006). As the bathymetric data set was collected at a time when the intermittent entrance at Shoalhaven Heads was closed, an additional data set of breached entrance conditions originating from 2015 was used to approximate the entrance conditions for the June 2016 storm event. Further information on the validation of the bathymetry interpolation can be found in the supplementary material of this manuscript.

Water level measurements at 15 min intervals for 5 tidal gauges (Fig. 1) were provided by OEH (distributed through Manly Hydraulics Laboratory). These measurements were already vertically referenced to AHD. Astronomical tide predictions based on harmonic analysis of one year of water level record (July 2015 to July 2016) were calculated using UTide Package for Matlab (Codiga, 2011). Non-tidal residuals were calculated by subtraction of the astronomical tide prediction from the observed water level (Fig. 2). Time-series of the Crookhaven Heads gauge measurements were used to force the models

ocean boundary (Fig. 3, upper plot). The peak water level of 1.65 m (AHD) at Crookhaven Heads was observed on 5 June 2016 at 21:00:00 LT. The gauges at Greenwell Point, Shoalhaven Heads, Nowra and Terara were used to validate the model performance.

Discharge measurements at 15 min intervals for the Shoalhaven River at Tallowa Dam were provided by New South Wales Water. This data is subject to uncertainty during peak discharge, because the discharge volume was too high to be recorded

by the measuring device. Therefore, the data was modified during several test simulations to enable the modelling of observed peak water levels at the upstream locations of Terara and Nowra. The discharge was estimated to peak with a maximum of approximately 3650 $m^3$ $s^{-1}$ (the device stopped recording at 2566 $m^3$ $s^{-1}$) on 6 June 2016 at 01:00:00 LT (Fig. 3, central plot). This modified data set was used to force the upstream boundary of the model.





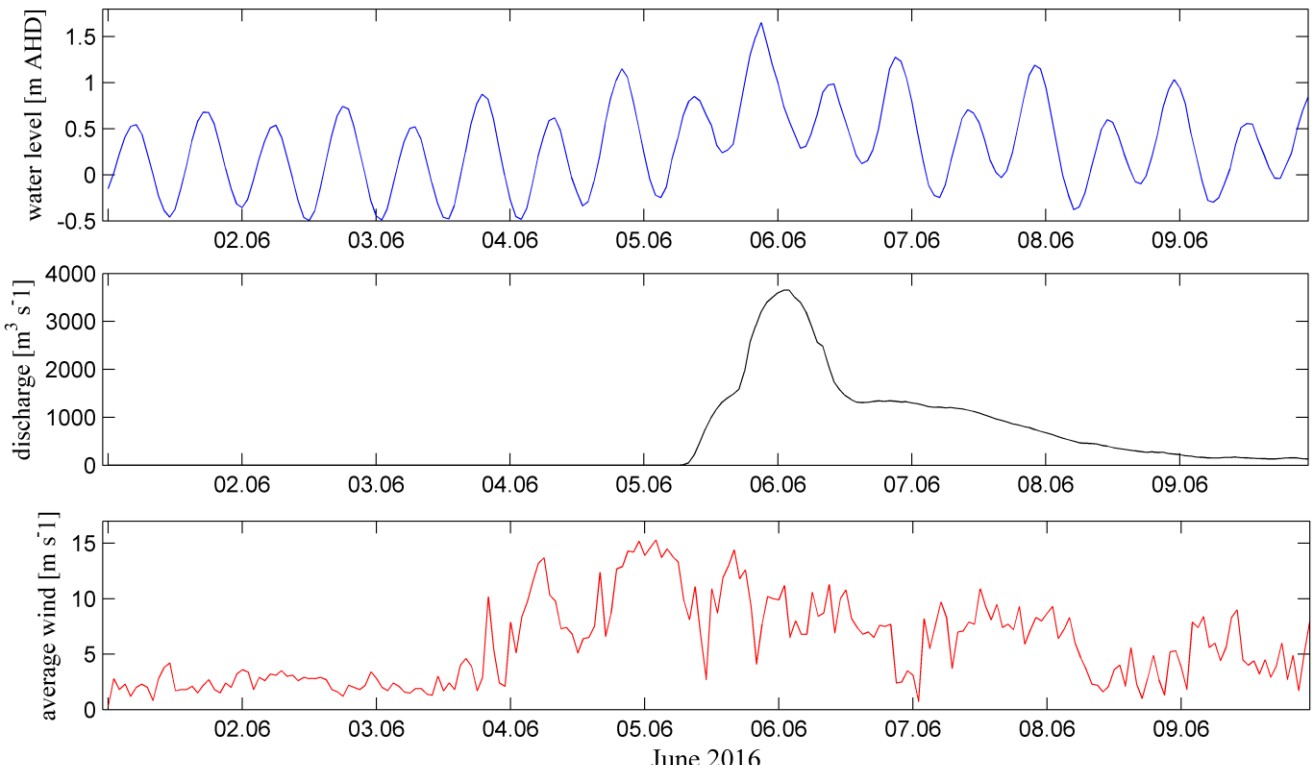

**Figure 3: Hydrological and meteorological model forcing taken from observational data of the June 2016 storm event. Water level measurements originate from Crookhaven Heads gauge, discharge measurements from Tallowa dam and wind measurements from Port Kembla.**

Wind data consisting of average wind speed, maximum gusts and average wind direction for Port Kembla were downloaded from the server of the Bureau of Meteorology (http://www.bom.gov.au/oceanography/projects/abslmp/data/). The average wind speed had a maximum of 15.2 m s$^{-1}$ (Fig. 3, lower plot) from an easterly direction.

The land use data were obtained from the NSW Department of Environment and Climate Change ((http://data.environment.nsw.gov.au/dataset/nsw-landuseac11c). They were used to create a file of spatially varying bottom
friction. This process is explained in more detail in Sect. 3.3.

### 3.2 Observational data

The area flooded during the June 2016 storm event was determined by using Sentinel-1 Synthetic Aperture Radar (SAR) imagery provided by Copernicus Sentinel Data, which was downloaded using the USGS Earth Explorer (https://earthexplorer.usgs.gov/). The imagery was taken on 6 June 2016 at 19:15:00 LT. Inundated areas were identified
through processing of the VH polarization band using the open source software SNAP toolbox (http://step.esa.int/main/download/).The SAR imagery was radiometrically calibrated, terrain corrected, speckle filtered and



reclassified based on the distribution of backscattering signals. This process is further illustrated in the supplementary material of this manuscript.

It was possible to separate the imagery into dry and inundated pixels based on the different reflectance of wet and dry areas. The resulting raster dataset of the observed flood extent was visually compared and adjusted using 75 aerial photographs of the flood extent. These photographs were taken during a helicopter survey on 6 June 2016 around 17:00:00 LT by the Shoalhaven City Council. Examples of photographs of the flood extent observed in the floodplain are shown in Fig. 4. Since several wetlands were not identified as inundated, Landsat 8 imagery downloaded from USGS Earth Explorer and taken on 6 June at 23:45:00 LT was used to further identify inundated areas and visually verify the SAR imagery reclassification. A band combination of 564 was shown to be most suitable to identify inundated areas.

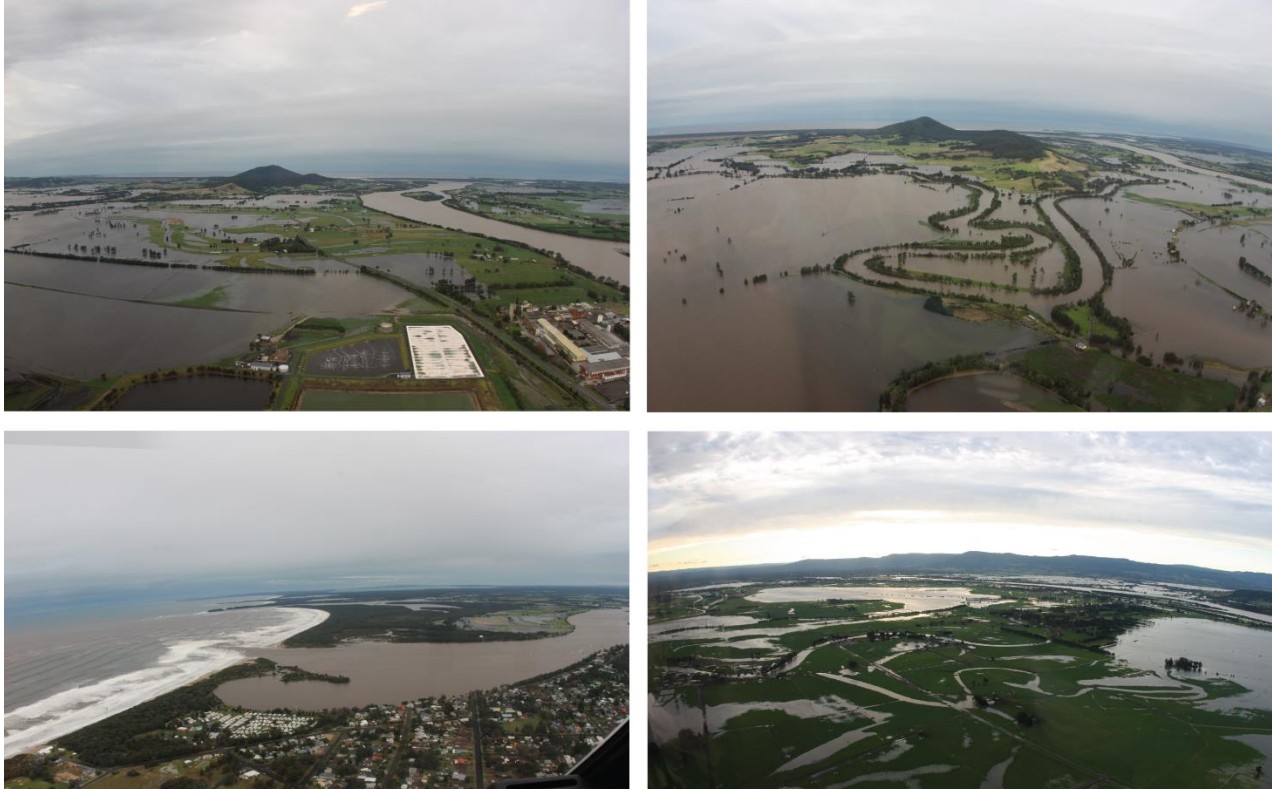

**Figure 4: Selection of aerial photographs taken by the Shoalhaven City Council on 6 June 2016 around 17:00:00 LT showing the flood extent of the June 2016 storm event. Looking from Nowra towards east into Broughton Creek floodplain (top left), looking from Broughton Creek floodplain towards east (top right), looking from Shoalhaven Heads towards south at the breached entrance and Comerong Island (bottom left) and looking from Greenwell Point towards northwest into Crookhaven floodplain.**

Aside from the water level measurements of tidal gauges mentioned before, measurements of two water level loggers (HOBO ® U20-001-04) at Comerong Island were used to validate the wetting and drying of the adjacent floodplain. Both



loggers were positioned in intertidal areas and measured water depths relative to their respective local elevation in 15 min intervals. The vertical accuracy of these measurements is reported to be 0.3 cm (Onset, 2017).

### 3.3 Hydrodynamic flood model

Maximum flood extents and inundation depths were simulated with the Delft3D model. The hydrodynamic numerical module Delft3D-Flow of the open source model Delft3D (Deltares, 2014) was used to simulate the resulting hydrodynamics using a combination of storm-tide and riverine discharge recorded for the June 2016 storm event. Delft3D-Flow has been shown to be capable of simulating processes relevant in coastal environments (Lesser et al., 2004). The finite difference model was carried out in a depth-averaged mode (2D) to solve the unsteady shallow water equations on a rectangular grid. The graphical user interface DelftDashboard (https://publicwiki.deltares.nl/display/DDB/Download) was used to create and pre-process input data for the Shoalhaven River model. The computational grid used Cartesian coordinates (GDA 1994 MGA Zone 56), had a horizontal resolution of 25 m and extended from the estuary's entrance at Crookhaven Heads upstream to the tidal limit at Burrier, where the bathymetric data coverage ends (Fig. 5). In consequence, the discharge measurements that originate from Tallowa dam were shifted approximately 25 km downstream to Burrier.







**Figure 5: Map showing the hydrodynamic model domain (grey outline), open boundaries in Crookhaven Heads and Shoalhaven Heads (bold red lines), the river discharge location upstream (orange dot) and monitoring points corresponding to tidal gauges (red dots) and water level loggers (green dots). The flood extent of the June 2016 storm event in the lower Shoalhaven floodplain is indicated in dark blue.**

The spatial discretization of the horizontal advection terms in the momentum equation was solved using the Delft3D flood scheme, which is recommended in the software manual for rapidly varying flows (Deltares, 2014). The parameter used to determine a suitable time step for the model simulations is the Courant-Friedrichs-Lewy number (CFL), which is defined by Eq. (1):

$$\text{CFL} = \Delta t \sqrt{\frac{gH}{\{\Delta x\}}} \tag{1}$$





where $\Delta t$ represents the time step in s, g the acceleration of gravity in m s$^{-1}$, H the total water depth in m and $\{\Delta x\}$ the horizontal grid spacing minimum in m (Deltares, 2014).

The ocean model boundary was forced with time-series of water level measurements taken at Crookhaven Heads gauge, whereas the upstream boundary was forced with time-series of discharge measurements taken at Tallowa dam. As the intermittent entrance in Shoalhaven Heads opened during the storm event, simulations using different boundary conditions were carried out (namely a one-open boundary setup with one boundary at Crookhaven Heads and a two-open boundary setup with an additional opening at Shoalhaven Heads). This was done to consider and compare the different entrance conditions observed during the storm event. Open boundaries were assumed to be spatially constant in terms of water level evolution along the boundary. Wind was applied spatially uniform due to the comparatively small model domain and data availability. Spatially varying bottom friction with respect to different land use types was defined using Manning's friction coefficients. Therefore, friction coefficients were taken from literature (Chow, 1959; Fisher and Dawson, 2011; Kaiser et al., 2011) and assigned to the land use data in a Geographic Information System. The threshold depth for the flooding of grid cells was set to 0.1 m. All other model parameters were kept to their default values. The simulation time of the storm event was set from 1 June 2016 00:00:00 LT to 7 June 2016 06:00:00 LT using a time step of 0.04 min. This long computational time was chosen to validate the model performance on a longer time scale than the actual storm event. All simulations were executed on a computer with an Intel Xeon E5-2670 processor with 12 cores and resulted in a computation time of 42 h.

The model was manually calibrated in an iterative manner. Simulations with a uniform bottom friction value were compared to ones with spatially varying friction coefficients in order to find the most suitable setup to replicate the observed water levels and flood extent of the June 2016 storm event. As mentioned in the input data section, the anomalous Shoalhaven River discharge data set was increased stepwise around flood peak. Furthermore simulations of varying threshold depths for flooding were compared.

After the model was calibrated for the June 2016 event, simulations using two-open boundaries including and excluding riverine discharge were carried out to quantify modelling differences in flood extent and inundation depth. The same simulations were also carried out using only one-open boundary at Crookhaven Heads to investigate how the modelling of flood extent is affected by different entrance conditions.

### 3.3.1 Verification and validation methods

The model performance was evaluated through the entire 7.25 day simulation period using statistical measures Eq. (2) and (3) as presented by Skinner et al. (2015):





$$R^2 = 1 - \frac{\sum (mi - oi)^2}{\sum (mi - meani)^2} \qquad (2)$$

$$E_{RMS} = [1/n \sum_{i=1}^{n} (mi - oi)^2]^{0.5} \qquad (3)$$

5   where $o_i$ represents the observed water level measurements at time step $_i$, $m_i$ the corresponding simulated water level at time $_i$, and $mean_i$ the mean of the simulated water levels.

In addition, peak water level differences were calculated by subtraction of the observed peak water level from the modelled peak water level.

The predictive quality of the model was quantified by the goodness of fit ($F$) measure Eq. (4) as presented by Bates et al. (2005).

$$F = \frac{FE_{obs} \cap FE_{mod}}{FE_{obs} \cup FE_{mod}} \qquad (4)$$

where $FE_{obs}$ and $FE_{mod}$ represent the observed and modelled flood extent.

Hence, the intersected area of observed and modelled flood extent is divided by the sum of both. The value of $F$ tends to 1 when the observed and modelled flood extent match exactly, and to zero when they don´t overlap at all. While Bates et al.
20   (2005) have defined good fit measurements for $F$-values greater than 0.5, Breihl et al. (2013) were more critical and set the threshold for good fit measures above 0.7. The $F$-value was calculated for the modelled flood extent on 6 June 2016 at 19:00:00 LT, because the SAR imagery was taken at this time.

Percentages of the model's correct estimations, overestimations and underestimations were derived through normalization of the three categories by the observed flood extent, as presented in Ramirez et al. (2016).

The validation of the flooding and drying processes was limited to two water level loggers at Comerong Island, which is located between the entrances of Crookhaven Heads and Shoalhaven Heads (Fig. 5). Modelled inundation depths were compared to observed ones by prior presented statistical measures of $R^2$ and $E_{RMS}$ as well as the difference in maximum inundation depth.

30   **3.3.2 Comparison of maximum flood extents and inundation depths**

Maps of maximum flood extent and maximum inundation depth of simulations including and excluding river discharge were derived from Delft3D output files using a GIS. Maximum flood extents were calculated for the one-open and two-open





boundary modelling setups. Maps of maximum inundation depth were only prepared for the two-open boundary setup as this was considered be more suitable for the replication of the June 2016 storm event. To visually enhance differences in inundation depth resulting from inclusion and exclusion of river discharge, the storm-tide only map was subtracted from the compound flooding map. These differences were further analysed through a reclassification of inundation depth pixels into

5      0.25 m intervals.

## 4. Results

### 4.1 Model performance and validation

Time series of modelled and observed water level for four tidal gauges in the Shoalhaven River are compared in Fig. 6. Modelled water levels are those resulting from a model forcing using storm-tide inputs at two-open boundaries and riverine

10    discharge at the upstream boundary. This model setup demonstrated to be the most suitable to reproduce the magnitude and timing of observed peak water levels at the four monitoring points. Statistical measures of $R^2$ and $E_{RMS}$ of 0.98 and 0.09 m for Greenwell Point, 0.98 and 0.14 m for Shoalhaven Heads, 0.99 and 0.15 m for Terara and 0.99 and 0.15 m for Nowra confirm this. The difference between modelled and observed peak water level was none for Greenwell Point and Nowra, -0.33 m for Shoalhaven Heads and 0.01 m for Terara.

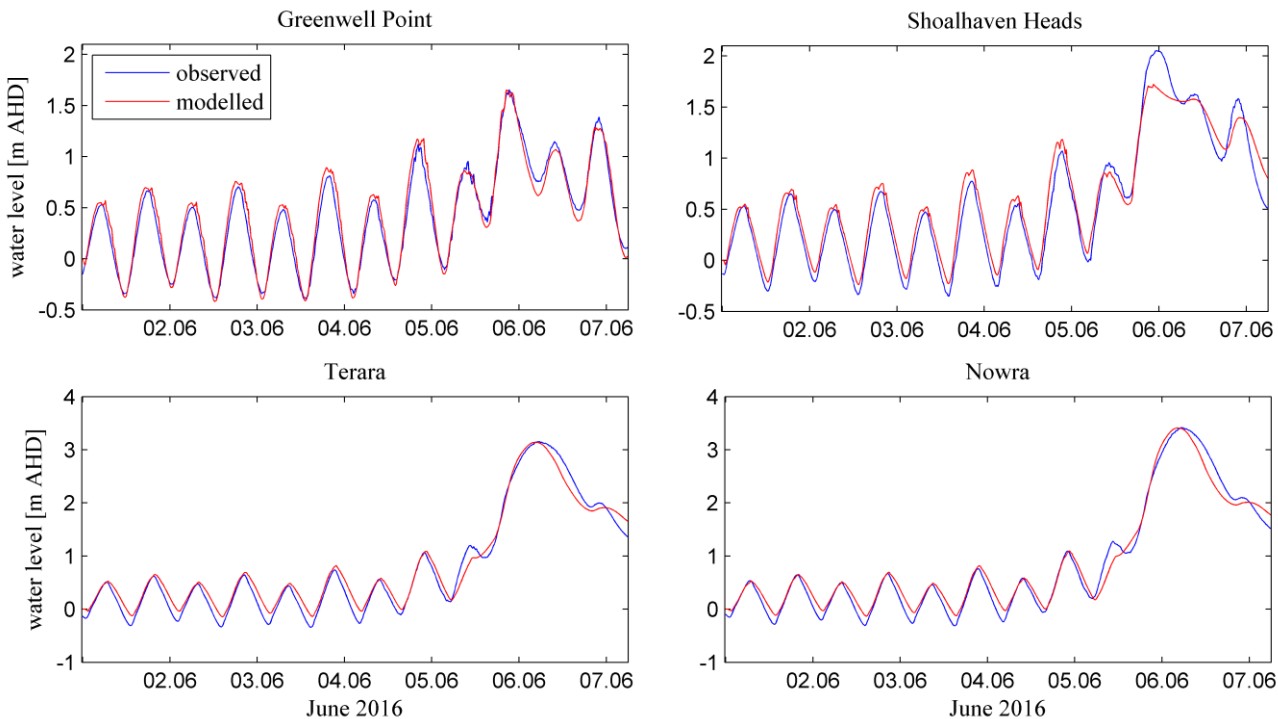

**Figure 6: Observed and modelled water levels of the June 2016 storm event for monitoring points in the Shoalhaven River. Modelled water levels result from a model forcing by storm-tide and river discharge using two-open boundaries.**





Time-series of modelled and observed inundation depth for two sites at Comerong Island are presented in Fig. 7. Modelled inundation depths were extracted from the same simulation as outlined above. The wetting and drying processes were reproduced reasonably well at the two water level logger sites. This is on one hand demonstrated by the minor overestimations in maximum inundation depth of 0.07 m at the northern site (WL1) and 0.04 m at the southern site (WL2),

and on the other hand, by the statistical measures of $R^2$ and $E_{RMS}$ of 0.93 and 0.61 m for WL1 and 0.90 and 0.53 m for WL2.

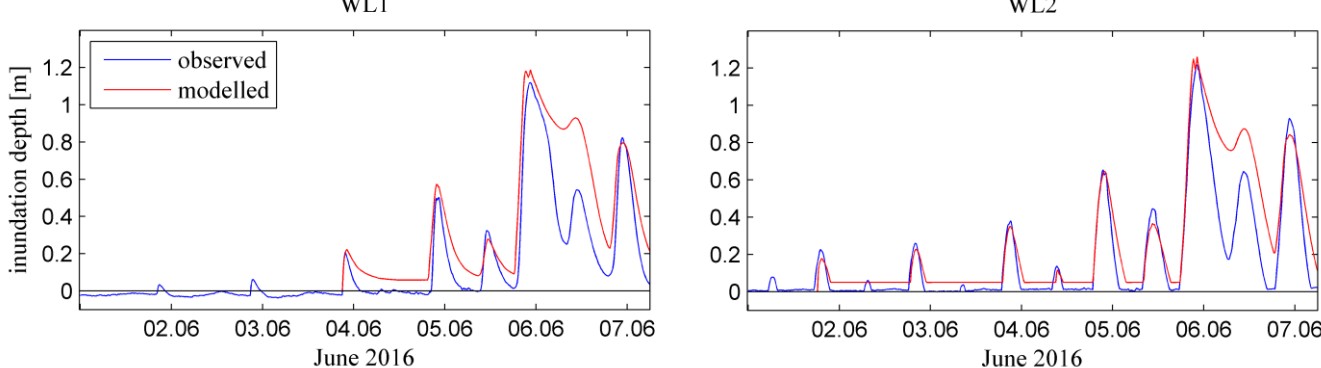

**Figure 7: Observed and modelled inundation depth of the June 2016 storm event for two water level loggers on Comerong Islands. Modelled inundation depths result from a model forcing by storm-tide and river discharge using two-open boundaries.**

The observed flood extent on 6th of June 2016 at 19:00h had a size of approximately 43.5 km². The model correctly

represented 89.7 % of this observed extent, which included most of the northern Broughton Creek floodplain and the largest patches of observed flooding in the southern Crookhaven floodplain (Fig. 8). Overestimations of modelled flooding were equal to 68 % of the observed flooding. Most of these overestimations were located in the Crookhaven floodplain surrounding Greenwell Point, as well as on Comerong and Kurrajong Islands. Underestimations of 10.2 % in modelled flood extent were located mainly in Brundee and Numbaa Swamp in the Crookhaven floodplain. The *F*-value calculated from

these results was equal to 0.53.

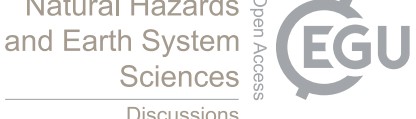



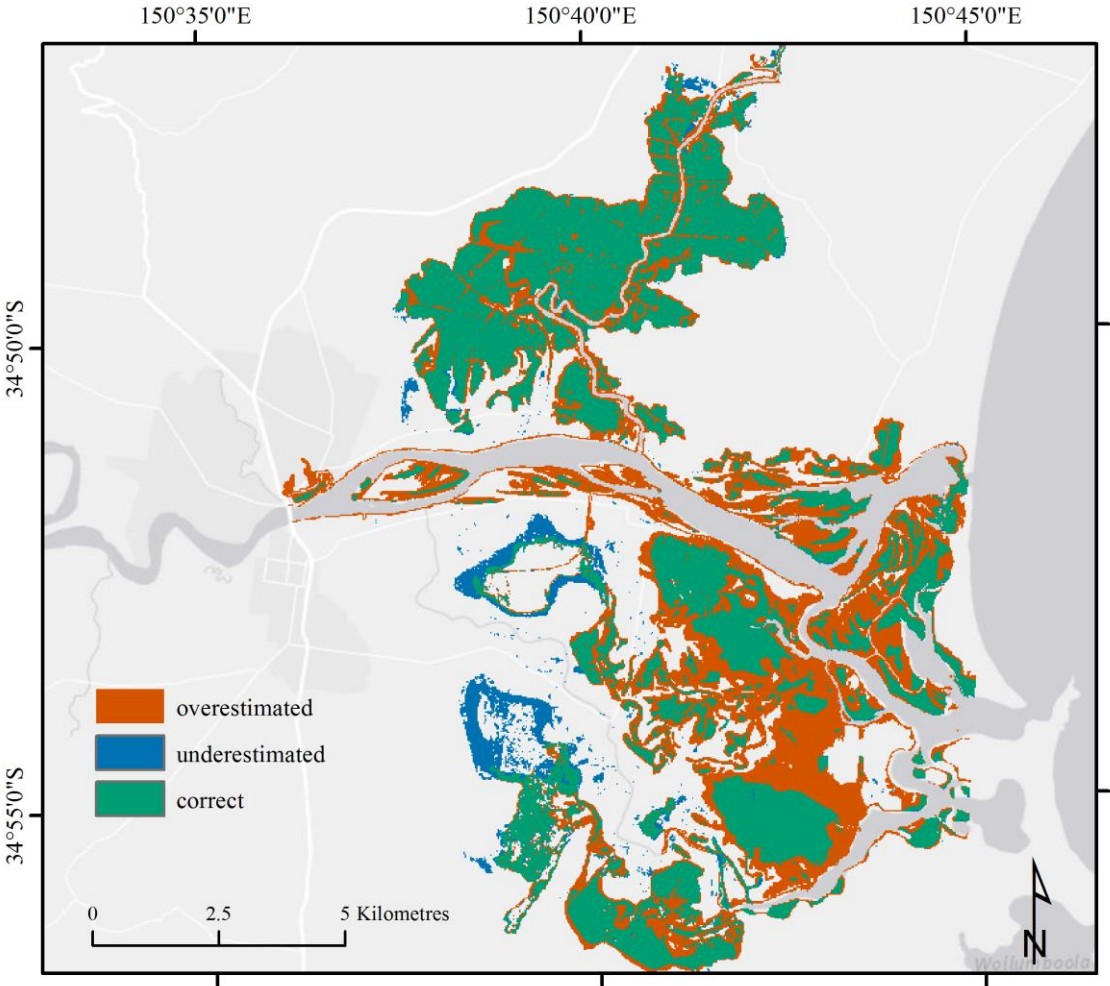

**Figure 8: Map showing locations correctly estimated, underestimated and overestimated by the model in the lower Shoalhaven floodplain.**

5  **4.2 Modelled water levels, flood extents and inundation depths**

Time-series of observed and modelled water levels for modelling setups using one- and two-open boundaries, including and excluding river discharge, at four monitoring points are presented in Fig. 9, whereas their corresponding flood extents are presented in Table 1.



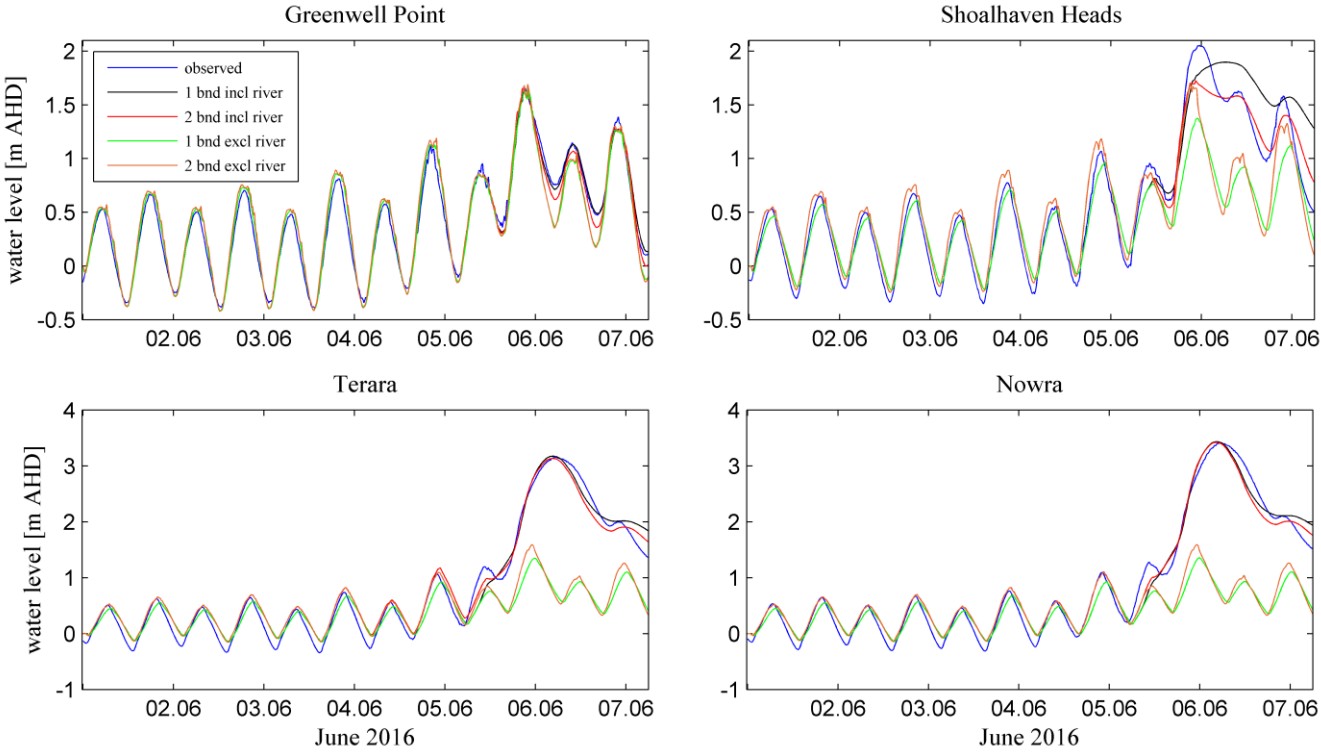

**Figure 9: Modelled and observed water levels for different model boundary forcing and entrance conditions.**

Comparison of modelling setups including and excluding river discharge demonstrated that peak water levels at Greenwell Point were only varying by a few centimetres. However, tidal low water following the flood peak was enhanced by approximately 0.25 m when river discharge was considered and demonstrated that this consideration is necessary to replicate the observed tidal low water (red line). The modelling using a closed boundary at Shoalhaven Heads further increased the modelled low water at Greenwell Point by 0.1 m (black line). The monitoring point at Shoalhaven Heads showed to be greatly influenced by the different modelling setups. The modelling setup using two-open boundaries including river discharge showed to be the best suitable setup to reproduce the observed water level at Shoalhaven Heads, even though the flood peak was underestimated by 0.33 m (see Sect. 4.1 and Fig. 9). A closure of the boundary at Shoalhaven Heads increased the water level by 0.17m, but also shifted the modelled flood peak from the observed (black line). The exclusion of river discharge caused an underestimation of the observed peak water level by 0.36 m in the two-open boundary setup (orange line) and an underestimation of 0.67 m in the one-open boundary setup (green line). However, none of the modelling setups was able to model the observed peak water level at Shoalhaven Heads, which appears to be enhanced by another driver. The modelling of different entrance conditions at Shoalhaven Heads revealed also a change in the tidal range for Greenwell Point and Shoalhaven Heads. The opening of the intermittent entrance increased the tidal range at Shoalhaven Heads by up to 0.25 m, whereas the tidal range at Greenwell Point increased by up to 0.08 m. The modelling of water levels at Terara and Nowra showed to be highly influenced by river discharge. The exclusion of river discharge caused an





underestimation of peak water levels by 1.59 m (two-open boundary) and 1.8 m (one-open boundary) at Terara and underestimations of 1.82 m (two open boundary) and 2.05 m (one-open boundary) at Nowra. The increase in tidal range due to an open boundary at Shoalhaven Heads were also present in Terara and Nowra (up to 0.3 m).

Changes in the modelled flood extent were large between the different modelling setups (Table 1). Percentage changes in
5   flood extent presented hereafter are in relation to the two-open boundary setup including river discharge. A closure of the open boundary at Shoalhaven Heads overall increased the modelled flood extent by 10.5 km² (equal to 15 %). The exclusion of river discharge decreased the flood extent by 19 km² (equal to 28 %) for the one-open boundary setup and by 20.5 km² (equal to 30 %) for the two-open boundary setup. Figure 10 presents these spatial differences in flood extent resulting from simulations including and excluding river discharge using the two-open boundary setup.

**Table 1: Flood extents resulting from different modelling setups.**

| Modelling Scenario | Flood extent (km²) |
|---|---|
| 1 bnd incl. river discharge | 78.1 |
| 2 bnd incl. river discharge | 67.6 |
| 1 bnd excl. river discharge | 48.6 |
| 2 bnd excl. river discharge | 47.2 |



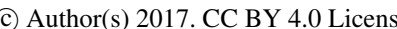

**Figure 10: Map of modelled maximum flood extents resulting from simulations including and excluding river discharge using the two-open boundary model setup. Flood extent of modelling including river discharge is indicated in blue, whereas the flood extent of modelling excluding river discharge is presented in orange.**

5   Orange areas in Fig. 10 are indicating the flood extent resulting from a model forcing by storm-tide water levels only, whereas blue areas display the flood extent resulting from a combination of storm-tide and river discharge inputs. Comparison of the modelled flood extents resulting from these simulations revealed spatial differences within the lower Shoalhaven floodplain. While the inclusion of river discharge caused only small changes in flood extent within the Crookhaven floodplain, the differences in the Broughton Creek floodplain were comparatively large. It appears that part of

10  the Shoalhaven River discharge entered Broughton Creek and enhanced the inundation of the surrounding floodplain. In contrast, the flood extent in areas closer to the ocean or the Crookhaven River changed only marginally when river discharge





was included. Figure 10 indicates a larger flood extent in the central part of the Crookhaven floodplain, but this difference is not as pronounced as the one observed in the Broughton Creek floodplain.

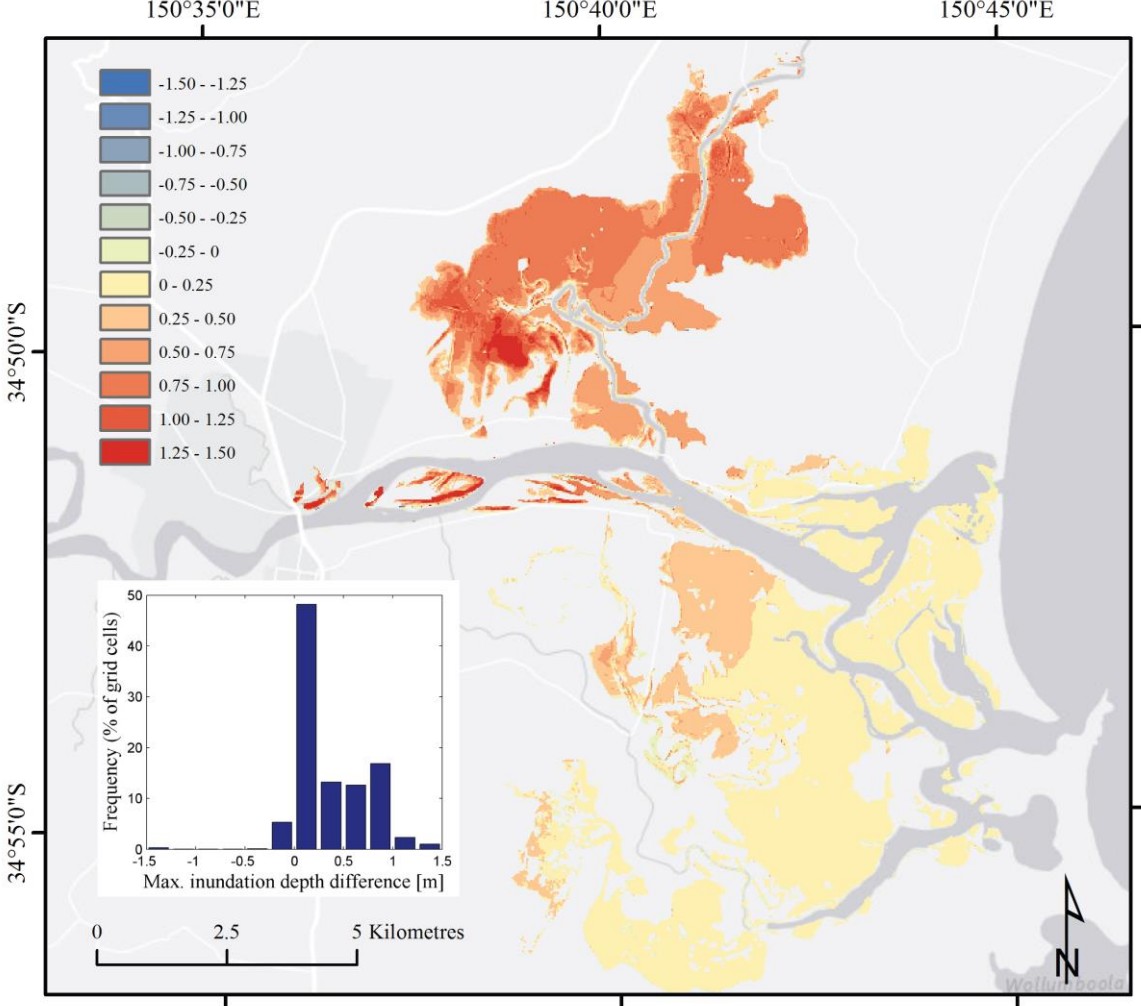

**Figure 11: Differences in maximum inundation depth in 0.25m intervals resulting from simulations including and excluding river discharge using the two-open boundary modelling setup. The histogram summarizes the percentage of grid cells falling into these intervals.**

Figure 11 presents differences in maximum inundation depths in 0.25 m intervals resulting from simulations including and excluding river discharge using the two-open boundary modelling setup. The largest differences in inundation depth of up to 1.5 m were observed around upstream locations such as Terara and Nowra, even though the flood extent was comparatively small in these areas. Most of the Broughton Creek floodplain was at least 0.5 m higher inundated when river discharge was included in the simulation, whereas local differences were as high as 1.5 m. For the remaining floodplain, differences in inundation depth were in the order of 0-0.25 m with some higher values in areas where the exclusion of discharge predicted no flooding at all. The histogram in Fig. 11 summarizes the percentages of grid cells falling into the 0.25 m intervals.



Approximately 48 % of flood extent grid cells had a difference of 0-0.25 m in inundation depth. Further the additional floodwater resulting from the inclusion of river discharge enhanced the water depth by 0.25-0.5 m for 13 % of grid cells, by 0.5-0.75 m for 12 % of grid cells and by 0.75-1.0 m for 17 % of grid cells. Very few locations also indicated lower inundation depths due to the consideration of river discharge (5.7 % of pixels). The mean difference in inundation depth for all grid cells was 0.34 m.

## 5. Discussion

### 5.1 Model performance

The underestimation of the observed peak water level at Shoalhaven Heads gauge likely results from the non-consideration of wave action at the intermittent entrance at Shoalhaven Heads.

The entrance naturally opened during the storm event and in consequence, the respective tidal gauge became also subject to wave-setup. Miller et al. (2006) reported that wave action can raise water levels at Shoalhaven Heads leading to an enhancement of the flood tide. This is possible once the entrance is opened and consistent with findings of Nielsen and Hanslow (1995) for wave-setup at other NSW river entrances. Similar influences of wave action on water levels at the mouth of an estuary were observed by Olbert et al. (2017) at the Lee River estuary in Ireland. The comparison of simulations including and excluding the Shoalhaven River discharge demonstrated no major differences in modelled water levels at Shoalhaven Heads and thus indicates that another driver must have enhanced the water level during the June 2016 event. The modelling of waves could answer this question, but our modelling of waves using the Delft3D Wave module was limited through model input data being not available.

The comparatively high $E_{RMS}$ observed during modelling of the inundation depth at Comerong Island is likely to be caused by the slowed down ebb flow. Inaccuracies in bottom friction values assigned to the complex wetland area and its channel network may explain the insufficient modelling of floodwater drainage. Following this assumption, the overestimations of the low and high water following the peak water level are likely the result of this undrained condition. The underestimation in flood extent in the Crookhaven floodplain may be the result of the spatial modelling resolution of 25 m. As the 5 m LiDAR elevation data is interpolated to the 25 m computational grid in order to reduce processing time, a loss of topographic detail is inevitable. In consequence, small creeks and channels that distribute floodwater may have been represented incorrect.

Overall, the prediction of water levels and flood extent as well as the representation of flooding and drying processes at Comerong Island demonstrated that the present model is able to replicate the main physical processes involved in the Shoalhaven Estuary during the June 2016 storm event. The SAR-imagery was shown to be a valuable source to determine the observed flood extent, even though it appeared to be limited in areas of inundation depths smaller than 0.25 m and saltmarsh/mangroves habitats due to their vegetation like reflectance. The limitation of remote sensing based flood detection



due to dense vegetation covers have also been observed by Teng et al. (2015). Nevertheless, the obtained model fit score of 0.53 demonstrates a high predictive skill of the present model and is similar to fit scores presented in Bates et al. (2005).

## 5.2 Interaction of storm-tide and riverine flooding mechanisms

The incorporation of riverine discharge demonstrated to be crucial for the replication of the June 2016 flood extent and water
levels in the lower Shoalhaven floodplain. Comparison of modelling results including and excluding riverine discharge clearly indicates that storm-tide and riverine flooding mechanisms were interacting and jointly causing the observed flooding patterns. While gauges at the entrance of the estuary such as Greenwell Point and Shoalhaven Heads appeared to be mainly controlled by storm-tide water levels and just marginally reacted to the model forcing by riverine discharge, upstream locations such as Nowra and Terara were shown to be highly influenced by the extreme discharge of the Shoalhaven River.
Furthermore, the comparison of simulations including and excluding river discharge indicated that the riverine flood wave was redirected into Broughton Creek and thus mostly affected the Broughton Creek floodplain. This was demonstrated on one hand by the large changes in flood extent surrounding Broughton Creek, and on the other hand, by the comparatively large differences in inundation depth in the Broughton Creek floodplain. Thereby, the timing of the riverine and storm-tide flood wave peaks may influence the exaggeration of extreme storm-tide water levels. While a coincidence of the two flood
waves at spring high tide may trap and pile up the riverine discharge, a coincidence at low tide may promote an efficient drainage of the riverine floodwater into the ocean. A blocking of riverine floodwater by a storm-tide was also observed by Chen et al. (2014) in the Tsengwen River. Spatial patterns of storm-tide and fluvial driven parts of a floodplain were also observed in other studies (Olbert et al., 2017) and likely apply to other estuaries exposed to compound flooding. The joint consideration appears to be important for the mid to upper reaches of the Shoalhaven River, whereas the lower floodplain
seems to be mainly controlled by coastal flooding mechanisms.

## 5.3 Modelling differences in flood extent and inundation depth

The comparison of the two-open boundary modelling setups including and excluding riverine discharge demonstrated large differences in modelled flood extent and inundation depth. The exclusion of riverine discharge caused a decrease of 20.5 km² in flood extent (30 %). In consequence, a separation of the two flood drivers in risk assessments may lead to large
underestimation of flood risk. This is also confirmed by the differences of up to 2 m between modelled and observed water levels at upstream gauges such as Nowra and Terara. Further the comparison of inundation depths between simulations including and excluding riverine discharge revealed an average increase in inundation depth of 0.34 m for the lower floodplain and local increases of up to 1.5 m in the Broughton Creek floodplain. The few locations indicating a lower inundation depth due to the consideration of river discharge are likely to be artefacts of data conversion processes. The
Delft3D outputs of maximum inundation depth were converted from vector into raster format, clipped by the outline of the maximum flood extent and finally intersected with each other. This was done using a 5 m pixel resolution and may have created these pixels of lower inundation depth (most of them are located at the waterfront).





The enhancement in inundation depth by riverine discharge has significant implications for risk estimations in monetary terms, because the flood damage to buildings is calculated as a function of inundation depth (Tsakiris, 2014; Jongmann et al., 2012). The additional riverine floodwater present during compound flooding may considerably change flood risk estimations for buildings in fluvial driven parts of a floodplain such as the Broughton Creek floodplain.

**5.4 Effects of changing entrance conditions**

The comparison of different entrance conditions deciphered that an opening of Shoalhaven Heads implies positive and negative effects, which should be considered when managing the intermittent entrance. Simulations using a closed boundary at Shoalhaven Heads showed a pile up of riverine floodwater during compound flooding conditions, which enhanced water levels at Shoalhaven Heads gauge and overall increased the modelled flood extent in the lower Shoalhaven floodplain. In

contrast, an opening of the Shoalhaven Heads entrance appeared to support the drainage of riverine floodwater. This is demonstrated by a decrease in modelled water levels and modelled flood extent. The downside of this floodwater drainage support is the additional exposure to storm-tide water levels and an exposure to wave action, which may further enhance water levels locally. Additionally, the exposure to swell waves is likely to cause erosion in this usually sheltered environment. Blacka and Coghlan (2017) documented such erosion after the June 2016 storm event at Shoalhaven Heads.

The modelling results further indicated an increase in tidal range of 0.27 m due to the opening of Shoalhaven Heads. It would not appear prudent to open the intermittent at Shoalhaven Heads, because multiple flooding drivers are acting together. A limitation of exposure to riverine flooding through an opening of Shoalhaven Heads appears to increase the tidal range as well as the exposure to storm-tides and wave action. As this exposure to marine flooding drivers is likely to increase in the context of climate change and sea-level rise (Hinkel et al., 2014; IPCC, 2014; Vitousek et al., 2017), it shouldn´t be

supported by permanently opening the entrance.

**6. Conclusion**

Our modelling results highlighted that not considering the interaction of different flooding mechanisms can lead to significant underestimation in flood risk. We recommend to consider storm-tide and riverine flooding drivers jointly when assessing coastal flood risk in estuaries. This is particularly important for estuaries with large catchment areas (> 10000

km², which are known to have a quick response time to extreme rainfall. Classification schemes (e.g. Roy et al. (2001)) and statistical analysis on the dependence of storm surges and extreme rainfall such as the one presented by Zheng et al. (2013) can guide which estuaries are subject to compound flooding.

To further examine the enhancement of extreme water levels by wave action at estuarine entrances, we recommend the use of a coupled wave-flow model (similar to the modelling presented in Olbert et al., 2017), even though model complexity and

computational times may considerably increase. However, the nature of the intermittent entrance at Shoalhaven Heads is




quite unique and the consideration of wave action in other estuaries may be less challenging due to almost static topobathymetric entrance conditions.

The analysis and comparison to other storm events would on one hand increase the validity of the presented modelling results, and on the other hand, further validate the present model. Zheng et al. (2013) have found the strongest dependence between storm surge and extreme rainfall at the east Australian coastline for storm events with a duration of 2-4 days. The expansion of the modelling can be challenging, because the availability of suitable data to validate hydrodynamic modelling results is known to be limited due to various reasons (Smith et al., 2012). To ensure that future storm events are recorded in a comprehensive manner, we recommend to collect observational data of storm events in an organized way similar to Haigh et al. (2015). The validation of flood hazard models is known to be still underdeveloped (Molinari et al., 2017). One option to overcome this demonstrated to be the use of water level loggers. Our model validation highlighted the potential to use this kind of data to verify the wetting and drying of intertidal areas. A detailed flood damage assessment using the presented uncertainties in flood extent and inundation depth would further enhance the presented results.

**Data availability.** Most of the used data sets, as well as the applied hydrodynamic modelling software, are freely available (web links are presented in Sect. 3). However, some data sets such as river discharge measurements, aerial photographs or water level logger measurements were provided on request by the institutions and colleagues mentioned in the acknowledgements.

**Author contributions.** KK, ATV and CDW designed the study. KK and RCC setup the model and ran the simulations. KK, RCC, ATV and CDW analysed and interpreted the results. KK wrote the paper with substantial input from all co-authors.

**Competing interests.** The authors declare that they have no conflict of interest.

**Acknowledgements.** KK expresses gratitude to the German Academic Exchange Service (DAAD) for supporting his MSc project with a scholarship. RCC and CDW were supported by Australian Research Council Linkage project LP130101025. The provision of the following data sets is highly appreciated: Bathymetric data of the Shoalhaven Estuary provided by NSW Office of Environment and Heritage; Water level data of tidal gauges at Crookhaven Heads, Shoalhaven Heads, Greenwell Point, Terara and Nowra provided by NSW Office of Environment and Heritage (distributed through Manly Hydraulics Laboratory); Land use data provided by NSW Department of Environment and Climate Change; LiDAR elevation data provided by Commonwealth of Australia (Geoscience Australia); Discharge measurements of the Shoalhaven River provided by Water NSW; Water level logger measurements provided by Kerrylee Rogers and Kirti Lal.; Sentinel-1 SAR-imagery provided by Copernicus Sentinel Data; Landsat-8 imagery provided by U.S. Geological Survey; Aerial photographs of the flood extent provided by Shoalhaven City Council.





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
