# Peer review of "Investigating compound flooding in an estuary using hydrodynamic modelling: A case study from the Shoalhaven River, Australia."

_Natural Hazards and Earth System Sciences, 2017_

## Referee Comment (RC1) · Anonymous Referee #1 · 7 Nov 2017

An excellent and well written study that is of clear importance and interest to the reader some extremely minor clarifications in the text needed (see below).

Abstract - a strong opening statement that is not entirely true, please consider starting "Many previous" Also, please consider modifying the line "we recommend to consider..." perhaps rephrase to "Therefore, joint probability analysis of storm-tide and riverine flooding is crucial in estuaries" Although true that quickly responding catchments are prone to compound hazard long-term duration events may also be an issue..

Other comments: p1 L27. Consider expanding the Zheng et al. 2013 reference to give some examples - such as Bangladesh (Lewis et al. 2013) p3 L5 Please consider

also Maskell et al. 2013 who found non-linear interaction effects to be small and that simplified hydrodynamic modelling techniques suitable for river-storm tide interaction in an idealised estuary p9 l1 - the method is similar to the water-line method to determine inundation area (e.g. Lewis et al. 2013b, perhaps consider adding this for clarity for the reader

REFS: Lewis, M., Bates, P., Horsburgh, K., Neal, J. and Schumann, G., 2013. A storm surge inundation model of the northern Bay of Bengal using publicly available data. Quarterly Journal of the Royal Meteorological Society, 139(671), pp.358-369.

Maskell, J., Horsburgh, K., Lewis, M. and Bates, P., 2013. Investigating River–Surge Interaction in Idealised Estuaries. Journal of Coastal Research, 30(2), pp.248-259.

Lewis, M., Schumann, G., Bates, P. and Horsburgh, K., 2013b. Understanding the variability of an extreme storm tide along a coastline. Estuarine, Coastal and Shelf Science, 123, pp.19-25.

---

## Referee Comment (RC2) · Anonymous Referee #2 · 23 Nov 2017

The manuscript presents a numerical model application to investigate the combined effect of river discharge and storm surge effects on flooding around the Shoalhaven river estuary (Australia). The manuscript is well presented (considering aspects below) and has the scientific merits for publication in NHESS. Although presenting a local study case, the model application is well conducted and results are interesting. However, prior to acceptance for publication, authors should address some aspects, listed below: - Abstract: First sentence of the abstract is exaggerated. Studies assessing estuarine processes and flooding certainly included both forcing conditions combined. - Clarify or change the term "entrance condition" (page 1, line 20; page 3, line 27). - Methods could be significantly shorter. In general, the manuscript is too long. Details

such as statistical methods for assessing model quality could be only cited. There is no need to present all the equations (2 to 4). Details of the CFL equation is also not needed (Eq. 1). In general, several details in the method section could be left out. Citations to some of the detailed aspects would be enough and will reduce the length of the manuscript. Details of computer processor, for example, are not needed (page 12, line 17). - Has the model been calibrated and validated against measured current velocity data? This is an important aspect that limits the reliability of the application. It would be important to present the calibration of the model for current velocities, even if this is done for a different period, when data is available. If not possible, this limitation should be mentioned in the manuscript. Estuarine modelling applications require an assessment of their capabilities to reproduce the estuarine hydrodynamics, not only water levels. - Aspects such as those discussed in page 21 (Model performance), could be better verified through a comparison of modelled and measured current velocities.

---

## Author Comment (AC1) · 4 Dec 2017

We appreciate the comments of the anonymous reviewer. Below, we respond to each of the points raised (reviewer's comments are included in quotation marks).

"Abstract - a strong opening statement that is not entirely true, please consider starting "Many previous" Also, please consider modifying the line "we recommend to consider..." perhaps rephrase to "Therefore, joint probability analysis of storm-tide and riverine flooding is crucial in estuaries" Although true that quickly responding catchments are prone to compound hazard long-term duration events may also be an issue.."

We agree with the reviewer's comments about our abstract and changed the opening statement to indicate and recognize that there are other studies which consider both flooding processes jointly. Furthermore, we have incorporated the suggested rephrasing of the abstract's recommendations. The importance of event duration is briefly addressed in the conclusion of the manuscript (p24 lines 3-5).

"p1 L27. Consider expanding the Zheng et al. 2013 reference to give some examples - such as Bangladesh (Lewis et al. 2013)"

The corresponding section was expanded by the suggested study of Lewis et al. (2013). The study demonstrated the enhancement of coastal extreme water levels by riverine discharge and therefore fits well in the context of our manuscript.

"p3 L5 Please consider also Maskell et al. 2013 who found non-linear interaction effects to be small and that simplified hydrodynamic modelling techniques suitable for riverstorm tide interaction in an idealised estuary"

The suggested reference (Maskell et al., 2013) was incorporated into the manuscript. It indeed provides a good example of compound flooding in an estuarine environment. We found however that the discussion of non-linear interaction effects between surge and river are not directly relevant to the section, also because these effects were found to be insignificant in determining the flood extent (and would also need to be addressed in length, which would render this part too long).

"p9 I1 - the method is similar to the water-line method to determine inundation area (e.g. Lewis et al. 2013b, perhaps consider adding this for clarity for the reader"

We thank the reviewer for pointing out the similarity of our flood extent determination using SAR data to methods presented in Lewis et al. (2013b). We integrated the reference into our methods section to improve clarity and guide the reader to similar work.

References: Lewis, M., Bates, P., Horsburgh, K., Neal, J. and Schumann, G., 2013. A

storm surge inundation model of the northern Bay of Bengal using publicly available data. Quarterly Journal of the Royal Meteorological Society, 139(671), pp.358-369. Maskell, J., Horsburgh, K., Lewis, M. and Bates, P., 2013. Investigating River–Surge Interaction in Idealised Estuaries. Journal of Coastal Research, 30(2), pp.248-259. Lewis, M., Schumann, G., Bates, P. and Horsburgh, K., 2013b. Understanding the variability of an extreme storm tide along a coastline. Estuarine, Coastal and Shelf Science, 123, pp.19-25.

СЗ

---

## Short Comment (SC1) · 21 Dec 2017

Comment to the paper: ***Investigating compound flooding in an estuary using hydrodynamic modelling: A case study from the Shoalhaven River, Australia.***

The authors investigate compound flooding via hydrodynamic modelling for an event occurred in 2016 in southeast Australia. I have only a few comments I would like to add to the discussion of this very interesting paper.

About the first sentence:
(1) *Previous modelling studies have considered storm-tide and riverine flooding independently,* (2) *even though joint probability analysis highlighted significant dependence between extreme rainfall and extreme storm surges in estuarine environments.*

(1) I agree with the other referee comments, that this sentence is exaggerated. Moreover, to get a more complete picture of the studies (which is already quite satisfying), you may add some additional references (the last two tackle the compound flooding modelling differently, i.e. via statistical modelling):

- van den Hurk B, van Meijgaard E, de Valk P, van Heeringen KJ, Gooijer J. Analysis of a compounding surge and precipitation event in the Netherlands. Environmental Research Letters. 2015 Feb 26;10(3):035001.
- Bevacqua, E., Maraun, D., Hobæk Haff, I., Widmann, M., and Vrac, M.: Multivariate statistical modelling of compound events via pair-copula constructions: analysis of floods in Ravenna (Italy), Hydrol. Earth Syst. Sci., 21, 2701-2723, https://doi.org/10.5194/hess-21-2701-2017, 2017.
- Van den Brink HW, Können GP, Opsteegh JD, Van Oldenborgh GJ, Burgers G. Estimating return periods of extreme events from ECMWF seasonal forecast ensembles. International Journal of Climatology. 2005 Aug 1;25(10):1345-54.

(2) the first sentence of the abstract and P2 l1: *even though joint-probability analysis highlighted significant dependence between extreme rainfall and extreme storm surges*
I suggest to state explicitly that even in places where no statistical dependence exist between the compound flooding drivers, there can still be risk of compound flooding (for example co-occurrence of astronomical high tide and extreme river discharge). In general, even when the statistical significance of the dependence between sea and river levels is null, it may happen to have co-occurrence of sea and river water level extremes.

P23 l24 *This is particularly important for estuaries with large catchment areas (> 10000 km²), which are known to have a quick response time to extreme rainfall. Classification schemes (e.g. Roy et al. (2001)) and statistical analysis on the dependence of storm surges and extreme rainfall such as the one presented by Zheng et al. (2013) can guide which estuaries are subject to compound flooding*
I suggest to discuss the part about large catchments taking into account what was previously argued by Klerk et al. (2015) (page 8, second column) in a paper about the dependence between compound flooding drivers in the Rhine-Meuse Delta: "*This makes the dependence found hardly relevant for policy making, as peaks*" (of sea level and river discharge) "*do not tend to arrive in the area of interest at the same time. In smaller water systems, these time lags between high water levels and discharges may be much smaller*" (due to quick catchment response time to extreme rainfall) "*such as found for the recent events in the North of the Netherlands as described in another contribution to this issue…*"

Klerk, W. J., Winsemius, H. C., Verseveld, W. J. V., Bakker, A. M. R., and Diermanse, F. L. M.: The co-incidence of storm surges and extreme discharges within the Rhine-Meuse Delta, Environ. Res. Lett., 10, 035005, https://doi.org/10.1088/1748- 9326/10/3/035005, 2015.

---

## Author Response (AR1)

We appreciate the comments of the two anonymous reviewers and one commenter relating to the underlying review of manuscript number nhess-2017-360. Please find below the author's replies (in blue colour) to each of these comments:

**5 Anonymous reviewer # 1 (AR1)**

Abstract - a strong opening statement that is not entirely true, please consider starting "Many previous" Also, please consider modifying the line "we recommend to consider..." perhaps rephrase to "Therefore, joint probability analysis of storm-tide and riverine flooding is crucial in estuaries" Although true that quickly responding catchments are prone to compound hazard long-term duration events may also be an issue..

We agree with the reviewer's comments about our abstract and changed the opening statement to indicate and recognize that there are other studies which consider both flooding processes jointly. Furthermore, we have incorporated the suggested rephrasing of the abstract's recommendations. The importance of event duration is briefly addressed in the conclusion of the

15 manuscript (p24 lines 3-5).

10

p1 L27. Consider expanding the Zheng et al. 2013 reference to give some examples - such as Bangladesh (Lewis et al. 2013)

The corresponding section was expanded by the suggested study of Lewis et al. (2013). The study demonstrated the enhancement of coastal extreme water levels by riverine discharge and therefore fits well in the context of our manuscript.

p3 L5 Please consider also Maskell et al. 2013 who found non-linear interaction effects to be small and that simplified hydrodynamic modelling techniques suitable for river-storm tide interaction in an idealised estuary

- 25 The suggested reference (Maskell et al., 2013) was incorporated into the manuscript. It indeed provides a good example of compound flooding in an estuarine environment. We found however that the discussion of non-linear interaction effects between surge and river are not directly relevant to the section, also because these effects were found to be insignificant in determining the flood extent (and would also need to be addressed in length, which would render this part too long).
- 30 p9 11 the method is similar to the water-line method to determine inundation area (e.g. Lewis et al. 2013b, perhaps consider adding this for clarity for the reader

We thank the reviewer for pointing out the similarity of our flood extent determination using SAR data to methods presented in Lewis et al. (2013b). We integrated the reference into our methods section to improve clarity and guide the reader to similar work.

5 References:

Lewis, M., Bates, P., Horsburgh, K., Neal, J. and Schumann, G.: A storm surge inundation model of the northern Bay of Bengal using publicly available data. Q. J. Royal Meteorol. Soc., 139, 358-369, doi: 10.1002/qj.2040, 2013a. Lewis, M., Schumann, G., Bates, P. and Horsburgh, K.: Understanding the variability of an extreme storm tide along a coastline. Estuar. Coast Shelf S., 123, 19-25, doi: https://doi.org/10.1016/j.ecss.2013.02.009, 2013b.

10 Maskell, J., Horsburgh, K., Lewis, M. and Bates, P.: Investigating River–Surge Interaction in Idealised Estuaries. J. Coastal Res., 30, 248-259, doi: https://doi.org/10.2112/JCOASTRES-D-12-00221.1, 2013.

**Anonymous reviewer # 2 (AR2)**

15 Abstract: First sentence of the abstract is exaggerated. Studies assessing estuarine processes and flooding certainly included both forcing conditions combined.

We agree with both reviewer's comments to our first sentence and therefore changed it (see also reply to first reviewer's comments above) to indicate and recognize that there are other studies which consider both flooding processes jointly.

20

Clarify or change the term "entrance condition" (page 1, line 20; page 3, line 27).

The term "entrance condition" was replaced by "boundary setups". From a modelling perspective it appears more useful to talk about different boundary setups at the intermittent entrance instead of simply "entrance conditions".

- 25 Accordingly, Page 1, line 20 now reads as follows: "In addition, comparison of different boundary setups at the intermittent entrance in Shoalhaven Heads indicated that a permanent opening, in order to reduce exposure to flooding, would increase tidal range and exposure to both storm-tide flooding and wave action." Page 3, line 27 now reads: "To quantify how changing boundary setups at the intermittent entrance in Shoalhaven Heads affect modelled water levels and flood extent."
- 30 Methods could be significantly shorter. In general, the manuscript is too long. Details such as statistical methods for assessing model quality could be only cited. There is no need to present all the equations (2 to 4). Details of the CFL equation is also not needed (Eq. 1). In general, several details in the method section could be left out. Citations to some of the detailed aspects would be enough and will reduce the length of the manuscript. Details of computer processor, for example, are not needed (page 12, line 17).

We agree with the comments of the reviewer. We have now shortened the methods section by removing unnecessary information (e.g. Eq. 1, processor characteristics etc.) as the reviewer suggested. Details of statistical equations (Eq. 2-4) were removed and addressed through citations. We have also moved information on the processing and validation of bathymetry

5 data to the supplementary material.

Has the model been calibrated and validated against measured current velocity data? This is an important aspect that limits the reliability of the application. It would be important to present the calibration of the model for current velocities, even if this is done for a different period, when data is available. If not possible, this limitation should be mentioned in the manuscript.
Estuarine modelling applications require an assessment of their capabilities to reproduce the estuarine hydrodynamics, not only water levels. Aspects such as those discussed in page 21 (Model performance), could be better verified through a comparison of modelled and measured current velocities.

The model hadn't been calibrated and validated against measured current velocities, but we recently received data for a different event. These measurements of current velocity were collected during neap tidal conditions on a day in September 2017. Therefore, we carried out an additional simulation of tidal conditions for the time period of data collection. Results of this simulation demonstrated a model underestimation of maximum current velocities by 1 cm s-1 (modelled = 0.116 m s-1, observed 0.122 m s-1). This comparison indicates that our model is able to replicate this hydrodynamic parameter quite well. We must note however that our comparison was limited to a single upstream location. We have now included this comparison

20 into our manuscript and also discussed its limitations. Specifically, in page 12, line 23, we have added: "In addition, the models ability to reproduce estuarine hydrodynamics was assessed by comparison of measured and modelled current velocities for a different event. Results show small model underestimation of maximum current velocities by 1 cm s-1."

In page 21, line 30, we have added: "The comparison of measured and modelled maximum current velocities demonstrated a

25 good reproduction of estuarine hydrodynamics. We must note however, that, due to limited availability of measured data, the comparison was restricted to a single location and a different event."

**Short Comment # 1 by Emanuele Bevacqua**

30

(1) Previous modelling studies have considered storm-tide and riverine flooding independently, (2) even though joint probability analysis highlighted significant dependence between extreme rainfall and extreme storm surges in estuarine environments.

(1) I agree with the other referee comments, that this sentence is exaggerated. Moreover, to get a more complete picture of the studies (which is already quite satisfying), you may add some additional references (the last two tackle the compound flooding modelling differently, i.e. via statistical modelling):

- van den Hurk B, van Meijgaard E, de Valk P, van Heeringen KJ, Gooijer J. Analysis of a compounding surge and precipitation event in the Netherlands. Environmental Research Letters. 2015 Feb 26;10(3):035001.

- Bevacqua, E., Maraun, D., Hobæk Haff, I., Widmann, M., and Vrac, M.: Multivariate statistical modelling of compound events via pair-copula constructions: analysis of floods in Ravenna (Italy), Hydrol. Earth Syst. Sci., 21, 2701-2723, https://doi.org/10.5194/hess-21-2701-2017, 2017.

Van den Brink HW, Können GP, Opsteegh JD, Van Oldenborgh GJ, Burgers G. Estimating return periods of extreme events
from ECMWF seasonal forecast ensembles. International Journal of Climatology. 2005 Aug 1;25(10):1345-54.

We agree with comments to our first sentence and therefore changed it (see also reply to first and second reviewer's comments above) to indicate and recognize that there are other studies which consider both flooding processes jointly. We appreciate the suggested additional references. All of these references present findings on compound flooding, but they are generally

15 statistical modelling studies rather than hydrodynamic modelling studies (even though Hurk et al. (2015) use regional climate model simulations). In consequence, a high-quality consideration of these studies would require more explanation (which would render this already quite long manuscript). Therefore, we refrain from discussing the suggested references.

(2) The first sentence of the abstract and P2 11: even though joint-probability analysis highlighted significant dependence

20 between extreme rainfall and extreme storm surges

I suggest to state explicitly that even in places where no statistical dependence exist between the compound flooding drivers, there can still be risk of compound flooding (for example co-occurrence of astronomical high tide and extreme river discharge). In general, even when the statistical significance of the dependence between sea and river levels is null, it may happen to have co-occurrence of sea and river water level extremes.

25

5

We appreciate these suggestions regarding statistical dependence between flooding drivers. However, the discussion of dependence and independence between drivers appears to fit more within the discussion of our manuscript rather than the abstract. Therefore, we have addressed the suggestions within our discussion section.

30 P23 l24 This is particularly important for estuaries with large catchment areas (> 10000 km2), which are known to have a quick response time to extreme rainfall. Classification schemes (e.g. Roy et al. (2001)) and statistical analysis on the dependence of storm surges and extreme rainfall such as the one presented by Zheng et al. (2013) can guide which estuaries are subject to compound flooding

I suggest to discuss the part about large catchments taking into account what was previously argued by Klerk et al. (2015) (page 8, second column) in a paper about the dependence between compound flooding drivers in the Rhine-Meuse Delta: "This makes the dependence found hardly relevant for policy making, as peaks" (of sea level and river discharge) "do not tend to arrive in the area of interest at the same time. In smaller water systems, these time lags between high water levels and discharges

5 may be much smaller"(due to quick catchment response time to extreme rainfall)"such as found for the recent events in the North of the Netherlands as described in another contribution to this issue..."

Klerk, W. J., Winsemius, H. C., Verseveld, W. J. V., Bakker, A. M. R., and Diermanse, F. L. M.: The co-incidence of storm surges and extreme discharges within the Rhine-Meuse Delta, Environ. Res. Lett., 10, 035005, https://doi.org/10.1088/1748-9326/10/3/035005, 2015.

10

We thank for the given suggestions. Indeed, the study of Klerk et al. (2015) is interesting to compare our findings to. The Rhine-Meuse Delta is characterized by a much larger catchment area than our study site (~ 170.000 km2 compared to 7.000 km2), and therefore displays a fairly large time lag between extreme sea-levels and discharges. Having this in mind, we have realized that term "large catchment area" appears inappropriate for our study site. However, in the context of east Australian

15 estuaries and rivers, a catchment area of 7.000 km2 and more is considerably large. We have now changed the according section to "Australian estuaries" in order to clarify our findings. Furthermore, we have incorporated the differences in lag time discussed by Klerk et al. (2015).

  - P3L6: Misspelling corrected.
  - P3L28: AR2 comment #2 regarding the term "entrance condition" has been addressed.
  - P6L21-26: Information has been moved into supplementary material to reduce the length of the manuscript and address AR2 comment #3.
  - P8L4-5: Information has been added to clarify the applied method and address AR1 comment #4.
  - P10L9-P11L7: Unnecessary information has been removed in order to reduce the length of the manuscript and address AR2 comment #3.
  - P11L28-30: AR2 comment #4 regarding validation of estuarine hydrodynamics has been addressed.
- P12L6-25: Details of statistical measures have been removed in order to reduce the length of the manuscript and address AR2 comment #3.
  - P20L29-31: AR2 comment #4 regarding validation of estuarine hydrodynamics has been addressed.
  - P22L26-28: Information has been added to clarify our findings and address the suggestions of Bevacqua comment #3.
- P22L30-31: Bevacqua comment #2 regarding the occurrence of compound flooding in places without statistical dependence between flood drivers has been addressed.
  - P23L27-28: Acknowledgements have been expanded to thank the reviewers for their comments.
  - P24L1-2: Acknowledgements have been expanded to thank for the provision of current velocity measurements.
  - P24L25-26: Missing references of Chow (1959) was added.
- P25/26: New reference of Klerk et al. (2015), Lewis et al. (2013a and 2013b) and Maskell et al. (2013) were added.
  - P27L31: Misspelling corrected.

[revised manuscript text omitted]